# Factors Associated with Emergency Department Visits and Consequent Hospitalization and Death in Korea Using a Population-Based National Health Database

**DOI:** 10.3390/healthcare10071324

**Published:** 2022-07-17

**Authors:** Junhee Park, Yohwan Yeo, Yonghoon Ji, Bongseong Kim, Kyungdo Han, Wonchul Cha, Meonghi Son, Hongjin Jeon, Jaehyun Park, Dongwook Shin

**Affiliations:** 1Department of Family Medicine & Supportive Care Center, Samsung Medical Center, School of Medicine, Sungkyunkwan University, Seoul 06351, Korea; junhee.park26@gmail.com (J.P.); chyhu888@naver.com (Y.J.); 2Department of Family Medicine, College of Medicine, Hallym University Dongtan Sacred Heart Hospital, Hwaseong 18450, Korea; 3Department of Statistics and Actuarial Science, Soongsil University, Seoul 06978, Korea; qhdtjd12@gmail.com (B.K.); hkd917@naver.com (K.H.); 4Department of Emergency Medicine, Samsung Medical Center, School of Medicine, Sungkyunkwan University, Seoul 06351, Korea; docchaster@gmail.com; 5Department of Pediatrics, Samsung Medical Center, School of Medicine, Sungkyunkwan University, Seoul 06351, Korea; meonghison@gmail.com; 6Department of Psychiatry, Samsung Medical Center, School of Medicine, Sungkyunkwan University, Seoul 06351, Korea; jhj001001@gmail.com; 7Center for Wireless and Population Health System, University of California, La Jolla, San Diego, CA 92093, USA; pjaehyun1014@gmail.com; 8Department of Clinical Research Design & Evaluation, Samsung Advanced Institute for Health Science & Technology (SAIHST), School of Medicine, Sungkyunkwan University, Seoul 06355, Korea

**Keywords:** emergency department, emergency department visit, hospitalization, hospital death, sociodemographic, health behavior

## Abstract

We aim to investigate common diagnoses and risk factors for emergency department (ED) visits as well as those for hospitalization and death after ED visits. This study describes the clinical course of ED visits by using the 2014–2015 population data retrieved from the National Health Insurance Service. Sociodemographic, medical, and behavioral factors were analyzed through multiple logistic regression. Older people were more likely to be hospitalized or to die after an ED visit, but younger people showed a higher risk for ED visits. Females were at a higher risk for ED visits, but males were at a higher risk for ED-associated hospitalization and death. Individuals in the highest quartile of income had a lower risk of ED death relative to lowest income level individuals. Disabilities, comorbidities, and medical issues, including previous ED visits or prior hospitalizations, were risk factors for all ED-related outcomes. Unhealthy behaviors, including current smoking, heavy alcohol consumption, and not engaging in regular exercise, were also significantly associated with ED visits, hospitalization, and death. Common diagnoses and risk factors for ED visits and post-visit hospitalization and death found in this study provide a perspective from which to establish health polices for the emergency medical care system.

## 1. Introduction

In Korea, rates of attendance in the emergency department (ED) have been increasing. The number of ED users in 2019 was 197.5 per 1000 subjects, an increase from 153.0 in 2004 according to the National Emergency Medical Center’s annual report [1]. The number of ED users increased by 44% from 7,074,378 in 2004 to 10,217,208 in 2019, but the number of emergency care facilities only increased about 8% in that time, from 481 in 2004 to 521 in 2019. The imbalance between ED capacity and use negatively affected the quality of medical care, patient satisfaction, and healthcare costs [2]. ED-related mortality also increased [3].

In order to provide appropriate medical care for patients with limited resources, the identification of the determinants related to ED visits [4] is essential. In previous studies, several sociodemographic, medical, and health behavior characteristics were proposed to be related with ED visits. Sociodemographic factors included age [5,6,7], sex [5,6,7], ethnicity [6], place of residency [5,6], income level [5,7], educational level [5], occupational class [5], and social deprivation level [6]. Medical factors included self-reported health status [5], diagnosis of multi-morbidities [6], having had a previous ED visit [7], and use of primary care [7]. Health behavior factors have rarely been reported. Smoking was observed to result in more frequent ED visits [6]. Some previous studies also investigated risk factors for hospitalization (ED hospitalization [8,9,10,11]) and mortality (ED death) occurring after ED visits [12,13]. Some of those risk factors were different from those associated with ED visits [8,9,10,11,12,13].

However, there are some conflicting results. Additionally, most of these studies were conducted in Western countries: the United States, England [6,10], Canada [5,7], and Australia [3,14]. A few studies investigated risk factors for ED use in Korea [8,13,15,16,17], but most of these did not have a population-based design. Therefore, study populations were relatively small, and generalizability to the entire Korean population was not possible. In addition, the availability of study data from medical records other than those related to the need for emergent care was limited as was individual health-related behavior data [8,13,16,17,18].

Therefore, in this study, we aim to investigate socioeconomic, medical, and behavioral risk factors for ED visitation, hospitalization, and death in Korea using a population-based national health database.

## 2. Materials and Methods

### 2.1. Study Design, Setting, and Population

This is a cross-sectional study using data from the Korean National Health Insurance Service (NHIS), a mandatory universal public health insurance system that covers the entire Korean population, except for Medicaid beneficiaries in the lowest income bracket (approximately 3% of the population). The NHIS database (DB) contains data on demographic factors of enrollees (e.g., age, sex, income level, and place of residence) and links the data to a death registry DB. Medical information (e.g., the International Classification of Disease 10th Amendment [ICD-10] ED visit primary diagnosis code and, based on medical expenditure claims and prescriptions, current and past medical treatments) was also collected in the NHIS claims DB.

In addition, the NHIS provides information from regular health examinations [19] for all beneficiaries and all employed individuals. This information includes self-administered health questionnaires on lifestyle factors (e.g., smoking status and history, alcohol consumption, and exercise habits); anthropometric measurements by clinical staff to include height, weight, blood pressure; and laboratory test results for glucose and lipids after overnight fasting [20,21].

In this study, we included subjects who underwent a health screening from 1 January 2014 to 31 December 2015. The data included those from a 50% randomly sampled population; the NHIS provided this sample because of a data size limitation according to its data provision policy. This study was approved by the Institutional Review Board of the Samsung Medical Center (IRB No SMC 2020-06-048).

### 2.2. Risk Factors: Sociodemographic, Medical, and Health Behavior Characteristics

An ED visit was an ED attendance by an individual in the claim DB within the study period. An ED hospitalization was defined as a hospitalization occurring immediately after an ED visit. ED death was defined as death within seven days after an ED visit.

### 2.3. Definition of ED Visit, ED Hospitalization, and ED Death

Sociodemographic variables included age (categorized into four groups: 19–39, 40–64, 65–74, and ≥75 years old), sex, urban or rural residence area, and household income level (into quintiles based on monthly insurance premium levels). The medical aid population, the poorest 3%, was merged into the lowest income quintile group.

Medical variables consisted of comorbidities and disabilities of patients, information from previous ED visits and hospitalizations, and body mass index (BMI). Comorbidities were identified from medical claims ICD-10 codes and relevant medication prescriptions within one year prior to health screening. Hypertension was identified by ICD-10 codes I10-I13 or I15 with the prescription of antihypertensive drugs; diabetes mellitus was identified by ICD-10 codes E11-E14 with prescription of anti-diabetic medications; dyslipidemia was identified by ICD-10 code E78 plus lipid-lowering medication; prescription; chronic obstructive pulmonary disease [COPD] was identified by ICD-10 codes J43-J44, except for J43.0; ischemic heart disease was identified by ICD-10 codes I21-I22, stroke by ICD-10 codes I63-I64, congestive heart failure [CHF] by ICD-10 code I50, and chronic kidney disease (CKD) by codes N18-N19; malignancy could be associated with any C-code; and depression was identified by ICD-10 codes F32-F33. Disability was registered in the qualification DB of the NHIS DB through the national registration system [22]. Previous ED visits and hospitalizations were defined as one or more visits or hospitalizations, respectively, within 1 year prior to the health screening date. In addition, BMI was calculated using weight (kg) divided by height in meters squared (m^2^), and individuals were classified into one of five groups: underweight (<18.5 kg/m^2^), normal (18.5–22.9 kg/m^2^), overweight (23–24.9 kg/m^2^), obese I (25–29.9 kg/m^2^), and obese II (≥30 kg/m^2^), according to Asia-Pacific guidelines from the Western Pacific Regional Office (WPRO) [23].

Cigarette smoking, alcohol consumption, and regular exercise behavior data were also collected. Smoking status was categorized into non-smokers, ex-smokers, and current smokers. Daily alcohol intake was calculated by multiplying the average frequency of alcohol intake (per week) and the typical number of standard drinks on each occasion [24]. Alcohol consumption was classified into three levels: none, mild (<30 g of alcohol/day), and heavy (≥30 g/day). Regular exercise was defined as more than 30 min of moderate physical activity at least five times per week or more than 20 min of strenuous physical activity at least three times per week [25].

### 2.4. Statistical Analysis

Categorical variables are presented as number and percentage, and continuous variables are presented as mean ± standard deviation (SD). Primary diagnosis was sorted by sex and age group in order to investigate common medical causes of ED visit. The associations between the independent variables and ED visit, ED hospitalization, and ED death are presented as odds ratios (ORs) and 95% confidence intervals (CIs) from multiple logistic regression. For those having multiple ED visits, the first episode was considered in the logistic regression. Statistical analyses were performed using SAS version 9.4 (SAS Institute Inc., Cary, NC, USA), and a *p*-value < 0.05 was considered statistically significant.

## 3. Results

### 3.1. General Characteristics of Study Participants

A total of 10,769,893 adults who had undergone health screening in 2014–2015 were enrolled in this study. Mean age of participants was 49.8 (standard deviation [SD] 14.5) and 49.0% were female. About 44.7% of the study population lived in urban areas. The most common underlying diseases were dyslipidemia (37.8%) and hypertension (37.1%), followed by diabetes mellitus (13.4%), COPD (6.1%), ischemic heart disease (5.4%), depression (5.2%), and CKD (4.0%). A total of 5.8% of the study population had a disability. A total of 689,323 individuals (6.4%) had a history of previous ED visits, and 191,793 individuals (1.8%) had a history of hospitalization within 1 year before the screening date. Mean BMI was 23.9 (SD 3.4). Approximately 21.7% of the study population were current smokers (*n* = 2,340,821), and 47.4% consumed alcohol in mild (*n* = 4,327,699 [40.2%]) or heavy (*n* = 775,535 [7.2%]) levels. Approximately 20.2% of subjects (*n* = 2,172,299) engaged in regular exercise (Table 1).

### 3.2. Common Primary Diagnosis for ED Visits

In order of frequency, the leading causes of ED visits were “gastroenteritis and colitis of infectious and unspecified origin” (A09), “open wound of wrist and hand” (S61), “abdominal and pelvic pain” (R10), “open wound of head” (S01), “dizziness” (R42), “calculus of kidney and ureter” (N20), “urticaria” (L50), “disorders of vestibular function” (H81), “intracranial injury” (S06), and “pain in throat and chest” (R07). The frequency of the diagnosis at ED visit of “gastroenteritis and colitis of infectious and unspecified origin” (A09) was 7.3% regardless of sex, followed by “Open wound of wrist and hand” (S61) was the diagnosis in 4.8% of visits, and “abdominal and pelvic pain” (R10) was the diagnosis for 4.4% at the time. In males, “open wound of head” (S01, 5.7%) was the third most common diagnosis. “Calculus of kidney and ureter” (N20, 3.1%) was also prominent in men as were “intracranial injury” (S06, 2.2%), “dizziness” (R42, 2.0%), and “urticaria” (L50, 2.0%). In females, symptoms related to neuro-vestibular function, including “dizziness” (R42, 3.8%), and “disorders of vestibular function” (H81, 2.8%) were the fourth and fifth most common reasons for ED visits, followed by “urticaria” (L50, 2.4%) and “gastritis and duodenitis” (K29, 2.2%) (Table 2). With regard to age, external injuries, including “open wound of wrist and hand” (S61) and “open wound of head” (S01), were less frequent causes of ED visits in older persons, while “cerebral infarction” (I63), “dizziness” (R42), and “disorders of vestibular function” (H81) were more frequent. In the over-75-years age group, “pneumonia” (J18, 3.04%) and “fracture of femur” (S72, 2.38%) were common primary diagnoses at ED visits (Table 3).

### 3.3. Prevalence of ED Visit, ED Hospitalization, and ED Death

Among study participants, 721,307 (6.7%) individuals visited EDs, 202,567 (1.9%) were admitted to the hospital after an ED visit, and 422 (0.004%) died within seven days after ED visit. The crude rates for ED visit, ED hospitalization, and ED death per 1000 persons at risk are presented in Figure 1. An increase in these rates by age is noticeable.

### 3.4. Risk Factors: Sociodemographic, Medical, and Health Behavior Characteristics

#### 3.4.1. Sociodemographic Factors

Table 4 shows the adjusted OR (aOR) for all listed characteristics. Interestingly, contrary to the crude rate according to age group, patients aged 65–74 years were at the lowest risk for ED visit (aOR: 0.70, 95% CI: 0.70–0.71) followed by the 40–64-year age group (aOR: 0.83, 95% CI: 0.82–0.83) and the ≥75 group (aOR: 0.83, 95% CI: 0.82–0.84). The reference age group was the 19–39-years group. However, the risk for ED hospitalization (aOR: 1.65, 95% CI: 1.62–1.69) and ED death (aOR: 34.54, 95% CI: 14.63–81.54) was higher in the older group. Women are at slightly higher risk for ED visit (aOR: 1.03, 95% CI: 1.02–1.03) than men, while the risk of ED hospitalization (aOR: 0.88, 95% CI: 0.87–0.89) and ED death (aOR: 0.45, 95% CI: 0.35–0.58) is much lower in women than in men. Persons who lived in urban areas showed a lower risk for ED visit (aOR: 0.90, 95% CI: 0.89–0.90) and ED hospitalization (aOR: 0.84, 95% CI: 0.83–0.85), while the risk for ED death (aOR: 0.98, 95% CI: 0.80–1.19) was not different from that of residents in rural areas. The risk for ED visit did not vary much among income level quintiles, but the risk of ED hospitalization (the highest quintile, aOR: 0.89, 95% CI 0.88–0.90) and ED death (the highest quintile, aOR: 0.73, 95% CI 0.55–0.95) was significantly higher in the lowest quintile group compared to the high-income group.

#### 3.4.2. Medical Factors

Subjects who had comorbidities (hypertension, diabetes mellitus, and chronic kidney disease) did not show an increased risk of ED visits, but these subjects did have a slightly higher risk of ED hospitalization and death. Persons with COPD, ischemic heart disease, stroke, CHF, and malignancy were at a higher risk for ED visit, hospitalization, and death. Persons with depression were at an increased risk for ED visit and hospitalization, but not for ED death. Persons with a disability were at an increased risk for ED visit (aOR: 1.24, 95% CI: 1.23–1.25), hospitalization (aOR: 1.31, 95% CI: 1.29–1.33), and death (aOR: 1.45, 95% CI: 1.14–1.85). Subjects who visited an ED within one year before study enrollment had a higher risk of ED visit than subjects who did not (aOR 2.51, 95% CI 2.50–2.53). Those who experienced an ED hospitalization within one year prior to study enrollment had a higher risk of ED hospitalization (aOR 2.57, 95% CI 2.52–2.61). The association between BMI and the risk of ED visit was an inverse one. Compared to the subjects with normal BMIs (18.5–22.9 kg/m2), the lean body group (the lowest quartile) was at an increased risk for ED visit (aOR: 1.18, 95% CI: 1.16–1.19), ED hospitalization (aOR: 1.37, 95% CI: 1.34–1.40), and ED death (aOR: 1.93, 95% CI: 1.35–2.74). A lower risk was observed in overweight and obese groups for the three study outcomes.

#### 3.4.3. Health Behavior Factors

Current smokers had a higher risk for ED visit (aOR: 1.14, 95% CI: 1.14–1.15) and ED hospitalization (aOR: 1.18, 95% CI: 1.17–1.20) than non-smokers, but there was no difference in ED death (aOR 1.01, 95% CI 0.75–1.37) (Table 4). Mild alcohol consumers were at lower risk for ED visit (aOR: 0.95, 95% CI: 0.95–0.96), ED hospitalization (aOR: 0.81, 95% CI: 0.80–0.82), and ED death (aOR: 0.80, 95% CI: 0.62–1.03) than non-drinkers. However, heavy alcohol consumers were at higher risk of ED visit (aOR: 1.11, 95% CI: 1.09–1.12) and ED death (aOR: 1.41, 95% CI: 0.96–2.06). Subjects who participated in regular exercise were at lower risk for ED visit (aOR: 0.94, 95% CI: 0.94–0.95), hospitalization (aOR: 0.87, 95% CI: 0.86–0.88), and death (aOR: 0.79, 95% CI: 0.61–1.03).

## 4. Discussion

This is the first comprehensive study exploring ED visits and clinical courses after ED visits, including hospitalization and mortality, in Korea. In our study, sociodemographic, medical, and health behavior characteristics of patients showed associations with ED visit, ED hospitalization, and ED death.

### 4.1. Reason for ED Visit

The most common causes for ED visits in Korea were “gastroenteritis and colitis of infectious and unspecified origin”, “abdominal and pelvic pain”, “open wound of wrist and hand” and “open wound of head”. These frequent diagnoses are consistent with the findings in studies from Canada [26], Australia [27], England [28], and other Asian countries [29,30,31]. However, there may have restrictions in entering the primary diagnosis code for the purpose of examination in order to prevent medical charge reduction. This would occur with a “cerebral infarction” (I63) diagnosis code for instance. Therefore, there is a limitation in that the accuracy of diagnosis could be lower in the ED. Among the common diagnoses, “mental health problems and drug abuse”, one of the leading causes of ED attendance in Western countries [32], was not observed in this study. Although the prevalence of depression in Korea has increased up to 5.3% and the use of antidepressants has ranged up to 38% [33], the prevalence is still lower than that in Western countries. In the United States, the ED visit rate for patients with mental health disorders was 52.9 per 1000 adults in 2017–2019 [34]. From statewide North Carolina Emergency Department data, 36,240 patients were transported by law enforcement; annual rates increased from 186.9 per 100,000 adult residents in 2009 to 279.2 in 2016 [35]. Among visits, the most common primary diagnoses were Mental Health Diagnoses (43.1%), and 20.4% resulted in hospital admission [35]. In Asian countries, mental health problems will be underestimated as the primary diagnosis in EDs because of a patient’s reluctance to seek psychiatric treatment due to the associated stigma. In Korea, the use of an injury or intoxication code (T code) instead of a psychiatric code (F code) is preferred. The lack of effective use of the Mental Health Act could worsen a patient’s condition and result in an increase in the number of cases requiring ED visits. Considering that the suicide rate in Korea was the highest among Organization for Economic Cooperation and Development (OECD) countries [36], further studies investigating medical facility use by persons with mental health problems and their related “accidents”, including suicide or drug abuse, are necessary.

### 4.2. Risk Factors: Sociodemographic, Medical, and Health Behavior Characteristics

#### 4.2.1. Sociodemographic Factors

While ED visits were more common in younger people, older patients were more likely to experience ED hospitalization or death. Since the national screening age is over 40, the study subjects under 40 years old may have had underlying disease or poor general health. In a similar study, among patients using the Diagnosis-Related Group (DRG) payment system, patients aged 19–44 years had a 1.15-fold higher risk of ED visits. Our finding is consistent with previous studies that suggested a greater complexity of managing the health and social care of older patients [10,12,13].

Women showed a higher OR for ED visit, but men had higher risk of ED hospitalization and death. The female sex predominance in ED-use frequency is controversial. Although women are reported to be more frequent users of EDs [7,15,32,37], several studies conducted in London, Canada, and South Korea found that men are better predictors for ED visits [5,6,16]. Since being male is often a risk factor for several chronic diseases, the odds for ED hospitalization and death might be higher than for women. This was demonstrated in previous studies [12,13,16,38,39,40].

ED visits and hospitalizations for subjects living in urban areas were lower than for those living in rural areas. However, there was no difference in ED death rate related to the area of residency. The accessibility to medical facilities in urban areas might be better than in rural areas, resulting in lower risk of ED visit or hospitalization for those in urban areas. However, some studies reported that an increase in primary healthcare use was associated with an increase in the use of emergency care [6]. Although the direct relationship between the accessibility to and the frequency of emergent care use is complicated, there was a definite disparity in ED death rate. After adjusting for income level, subjects residing in rural areas had a significantly higher ED death rate than those residing in urban areas. According to the national territorial monitoring report, the average accessibility to general hospitals nationwide is 20.9 km with a maximum of 96.8 km. Therefore, 75.3% of the population can travel to a general hospital within 10 min (5 km) from their residence [41]. However, 2.11 million people did not arrive at the general hospital within 20 min [41]. In order to reduce the medical disparity between regions, significant effort went into prioritizing resources toward those in large population areas that cannot reach general hospitals within 20 min [42].

Subjects with higher incomes had a lower risk of ED hospitalization and death than those with lower incomes. ED visits were similar between the income level groups. Low socioeconomic status is an important determinant of health status [43], but the financial barriers observed with primary care were not observed with ED visits [44]. Our study findings suggest that non-financial barriers, such as disability, comorbidities, and smoking status, were more influential than the additional costs of ED visits with Korea’s health coverage policies. In accordance with the Emergency Medical Service Act, article 23, a non-emergency patient bears the payment of medical services fees, approximately 60,000 won [45]. The higher ORs for ED hospitalization and ED death for those in the low-income bracket can be explained by their lack of optimal health management, such as the lack of access to regular medical examinations and ambulatory care services [46,47].

#### 4.2.2. Medical Factors

Comorbidities were major risk factors for ED visit, hospitalization, and death in this study. The burden of multi-morbidity was the strongest clinical predictor of ED attendance in previous studies [6,10]. This attendance can be caused by an acute exacerbation of chronic disease, such as infection or acute complication. Efforts to prevent ED visits of those with comorbidities are needed.

In addition, those with disabilities were at higher risk of ED visit, hospitalization, and death. In Korea, people with disabilities are more likely to be hospitalized for Ambulatory Care Sensitive Conditions (ACSCs) due to their lower accessibility to primary care [48]. In a systematic review, higher accessibility to the primary healthcare system was associated with fewer ACSCs admissions after adjustment for individual health status [49]. Continuity of care, the availability to contact one’s own primary care physician, was associated with lower rates of ED admission [10] as well as CVD incidence and mortality [50,51]. Our study findings consistently suggest that strengthening primary care for those who have comorbidities and/or disabilities will prevent health deterioration to the point of requiring emergency management.

Those with previous ED visits and hospitalizations were at high risk of ED visit, hospitalization, and death independent of other sociodemographic, medical, and behavioral factors in our study. A previous study from a single center ED revealed that frequent ED users were more likely to be in poorer health, older, or have a chronic disease or a mental health disorder than occasional ED users [18]. Therefore, post-discharge care for ED attenders should be suggested to reduce frequent use of emergency care and to protect the capacity of the ED (i.e., paramedic-delivered care transitions intervention [52,53]).

The probability of ED visit, hospitalization, and death were higher among underweight subjects. Lower body weight is a risk factor for CKD [54,55], COPD^34^, and infectious diseases, including pulmonary tuberculosis [56], which can result in frequent ED visits due to aggravation of health status. While body weight and poor health conditions may have a reverse causality, adequate nutrition and maintaining appropriate body weight is important for ensuring healthy conditions in the context of emergent care regardless of the presence of underlying diseases.

#### 4.2.3. Health Behavior Factors

In the present study, former smokers were at a lower risk of ED hospitalization and ED death than current smokers. Smoking cessation has health policy implications not only in terms of ambulatory or inpatient care, but also for emergency care. Previous studies demonstrated significant reductions in mortality, readmissions, and ED use after hospital-initiated smoking cessation was implemented [57]. Indeed, the majority of ED patients who currently smoke were willing to participate in ED-initiated interventions for smoking cessation [58,59]. Clinicians in EDs are in the best position to recommend smoking cessation for those who present to EDs with smoking-related diseases.

As expected, heavy alcohol consumption was significantly associated with more frequent ER visits. However, mild alcohol consumers had the lowest risk for ED visits, hospitalizations, and death. Mild alcohol consumers usually have relatively healthy behaviors and conditions, which result in a J-shaped pattern in the risk of CVD incidence according to drinking level [60]. Consistent findings that occasional light-to-moderate drinkers had a lower risk of ED visits compared to ex-drinkers and heavy drinkers were also reported [61].

In addition, persons engaging in regular exercise also showed lower probability for ED visits, hospitalizations, and deaths. In a previous study, when supervised physical exercise was applied to sedentary older individuals, reductions in ED visits, ED hospitalizations, and length of hospital stays were observed [62]. A comprehensive approach to reduce modifiable risk factors (e.g., smoking cessation, avoidance heavy alcohol consumption, and engagement in regular exercise) identified in this study would be worthy when considering the burdens on both primary and emergent care.

### 4.3. Limitaions

There are several limitations in this study. Because health screening participants are generally healthy and exhibit relatively healthier behaviors in terms of smoking, drinking, and exercise than non-participants, selection bias may have occurred. However, our study population covered about 75% of all eligible adults in Korea. Second, given the retrospective study design, some possible confounding factors (i.e., environmental factors or psychological factors) that may contribute to patients’ decisions to attend EDs may have been missed because information was unavailable. Third, there was the potential for under-recording of underlying comorbidities, and information on severity of comorbidities was not available. Therefore, our consideration of comorbidities may not have been optimal.

## 5. Conclusions

Our study identified comorbidities, disabilities, being underweight, previous ER visits and/or hospitalizations, smoking, heavy alcohol consumption, and physical inactivity as risk factors that contributed to ED visits, ED hospitalizations, and ED deaths in Korea. The evidence provided in this study will inform future studies and provides a perspective for establishing health polices for emergency medical care in Korea.

## Figures and Tables

**Figure 1 healthcare-10-01324-f001:**
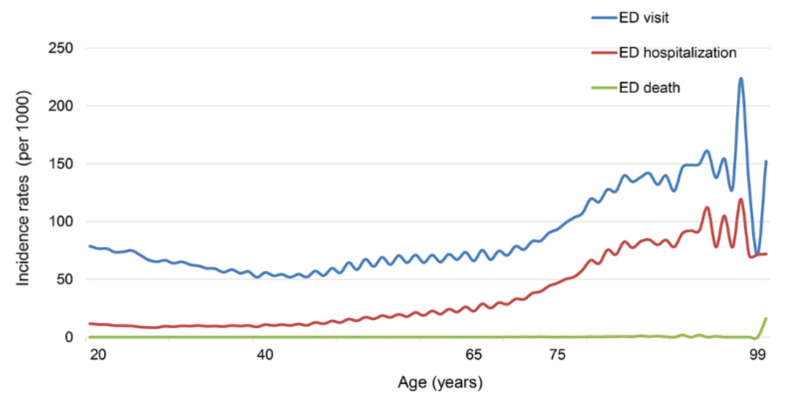
Prevalence of ED visit, ED hospitalization, and ED death.

**Table 1 healthcare-10-01324-t001:** Sociodemographic, medical, and health behavior characteristics of the study population.

	*n*	%
Total	10,769,893	100
**Sociodemographic**		
Age, mean (SD), years	49.8 (14.5)	
19–39	2,641,530	24.5
40–64	6,341,575	58.9
65–74	1,275,703	11.9
≥75	511,085	4.8
Sex		
Female	5,279,111	49.0
Place of residence		
Urban	4,818,898	44.7
Household income		
Q 1+ (lowest)	2,076,085	19.3
Q 2	2,375,031	22.1
Q 3	2,257,015	21.0
Q 4	2,085,322	19.4
Q 5 (highest)	1,976,440	18.4
**Medical**		
Comorbidities		
Hypertension	3,990,704	37.1
Diabetes mellitus	1,441,082	13.4
Dyslipidemia	4,069,851	37.8
Chronic obstructive pulmonary disease	658,692	6.1
Ischemic heart disease	579,933	5.4
Stroke	165,203	1.5
Congestive heart failure	140,445	1.3
Chronic kidney disease	427,938	4.0
Malignancy	244,305	2.3
Depression	556,613	5.2
Disability	625,631	5.8
Previous ED visit		
None	10,080,570	93.6
≥1	689,323	6.4
Previous hospitalization		
None	10,578,100	98.2
≥1	191,793	1.8
BMI, mean (SD), kg/m^2^	23.9 (3.4)	
<18.5	400,318	3.7
18.5–22.9	4,121,556	38.3
23–24.9	2,600,660	24.2
25–29.9	3,165,553	29.4
≥30	481,806	4.5
**Behavioral**		
Smoking status		
Never smoked	6,703,159	62.2
Ex-smoker	1,725,913	16.0
Current smoker	2,340,821	21.7
Alcohol drinking status		
Non-drinker	5,666,659	52.6
Mild drinker	4,327,699	40.2
Heavy drinker	775,535	7.2
Regular exercise	2,172,299	20.2

Abbreviations: SD = standard deviation, Q = quartile, ED = emergency department, BMI = body mass index.

**Table 2 healthcare-10-01324-t002:** Ten most common primary diagnoses for visits to the emergency department according to sex.

Total			Male			Female		
Diagnosis, ICD-10	*n*	%	Diagnosis, ICD-10	*n*	%	Diagnosis, ICD-10	*n*	%
A09 (gastroenteritis and colitis of infectious and unspecified origin)	52,711	7.3	A09 (gastroenteritis and colitis of infectious and unspecified origin)	21,658	5.8	A09 (gastroenteritis and colitis of infectious and unspecified origin)	31,053	8.9
S61 (open wound of wrist and hand)	34,919	4.8	S61 (open wound of wrist and hand)	21,327	5.8	R10 (abdominal and pelvic pain)	18,823	5.4
R10 (abdominal and pelvic pain)	32,023	4.4	S01 (open wound of head)	21,173	5.7	S61 (open wound of wrist and hand)	13,592	3.9
S01 (open wound of head)	28,645	4.0	R10 (abdominal and pelvic pain)	13,200	3.6	R42 (dizziness)	13,230	3.8
R42 (dizziness)	20,716	2.9	N20 (calculus of kidney and ureter)	11,389	3.1	H81 (disorders of vestibular function)	9766	2.8
N20 (calculus of kidney and ureter)	16,581	2.3	S06 (intracranial injury)	8155	2.2	L50 (urticaria)	8303	2.4
L50 (urticaria)	15,760	2.2	R42 (dizziness)	7486	2.0	K29 (gastritis and duodenitis)	7736	2.2
H81 (disorders of vestibular function)	14,862	2.1	L50 (urticaria)	7457	2.0	S01 (open wound of head)	7472	2.1
S06 (intracranial injury)	14,790	2.1	R07 (pain in throat and chest)	6959	1.9	S06 (intracranial injury)	6635	1.9
R07 (pain in throat and chest)	12,454	1.7	H81 (disorders of vestibular function)	5096	1.4	R51 (headache)	5579	1.6

Abbreviations: ICD-10 = International Classification of Disease, 10th Amendment code.

**Table 3 healthcare-10-01324-t003:** Ten most common primary diagnoses for visits to the emergency department according to age.

Age (19–39)			Age (40–64)			Age (65–74)			Age (≥75)		
Diagnosis, ICD-10	*n*	%	Diagnosis, ICD-10	*n*	%	Diagnosis, ICD-10	*n*	%	Diagnosis, ICD-10	*n*	%
A09 (gastroenteritis and colitis of infectious and unspecified origin)	18,279	11.1	A09 (gastroenteritis and colitis of infectious and unspecified origin)	27,061	6.8	A09 (gastroenteritis and colitis of infectious and unspecified origin)	4963	5.0	A09 (gastroenteritis and colitis of infectious and unspecified origin)	2408	4.2
S61 (open wound of wrist and hand)	10,541	6.4	S61 (open wound of wrist and hand)	21,086	5.3	R42 (dizziness)	4501	4.5	I63 (cerebral infarction)	2295	4.0
R10 (abdominal and pelvic pain)	9621	5.8	R10 (abdominal and pelvic pain)	17,481	4.4	R10 (abdominal and pelvic pain)	3333	3.3	R42 (dizziness)	2290	4.0
S01 (open wound of head)	6835	4.1	S01 (open wound of head)	17,007	4.3	S01 (open wound of head)	3274	3.3	J18 (pneumonia)	1754	3.0
L50 (urticaria)	4136	2.5	R42 (dizziness)	12,128	3.1	H81 (disorders of vestibular function)	3165	3.2	R10 (abdominal and pelvic pain)	1588	2.8
K29 (gastritis and duodenitis)	3745	2.3	N20 (calculus of kidney and ureter)	11,257	2.8	I63 (cerebral infarction)	2717	2.7	S01 (open wound of head)	1529	2.7
S93 (dislocation, sprain, and strain of joints and ligaments at ankle and foot levels)	3605	2.2	L50 (urticaria)	9907	2.5	S06 (intracranial injury)	2670	2.7	S06 (intracranial injury)	1502	2.6
N20 (calculus of kidney and ureter)	3402	2.1	H81 (disorders of vestibular function)	9082	2.3	S61 (open wound of wrist and hand)	2567	2.6	H81 (disorders of vestibular function)	1499	2.6
R50 (fever of other and unknown origin)	3216	2.0	S06 (intracranial injury)	8318	2.1	R07 (pain in throat and chest)	1878	1.9	S72 (fracture of femur)	1371	2.4
K52 (other noninfective gastroenteritis and colitis)	2315	1.4	R07 (pain in throat and chest)	7827	2.0	I20 (angina pectoris)	1758	1.8	K80 (cholelithiasis)	896	1.6

Abbreviations: ICD-10 = International Classification of Disease, 10th Amendment code.

**Table 4 healthcare-10-01324-t004:** Risk factors for ED visit, ED hospitalization, and ED death.

	ED Visit		ED Hospitalization		ED Death	
Variables	*n* = 721307	aOR * (95% CI)	*n* = 202567	aOR * (95% CI)	*n* = 422	aOR * (95% CI)
Age, years						
19–39	165,016	1 (Ref.)	25,088	1 (Ref.)	6	1 (Ref.)
40–64	398,263	0.83 (0.82, 0.83)	103,511	1.17 (1.15, 1.19)	151	8.34 (3.65, 19.06)
65–74	100,326	0.70 (0.70, 0.71)	42,555	1.23 (1.21, 1.26)	106	14.55 (6.19, 34.21)
≥75	57,702	0.83 (0.82, 0.84)	31,413	1.65 (1.62, 1.69)	159	34.54 (14.63, 81.54)
Female (vs. male)	350,331	1.03 (1.02, 1.04)	94,878	0.88 (0.87, 0.89)	137	0.45 (0.35, 0.58)
Urban (vs. rural)	297,932	0.90 (0.89, 0.90)	77,517	0.84 (0.83, 0.85)	162	0.98 (0.80, 1.19)
Household income						
Q 1 + (lowest)	146,924	1 (Ref.)	45,288	1 (Ref.)	109	1 (Ref.)
Q 2	151,845	0.99 (0.97, 0.99)	37,452	0.92 (0.91, 0.93)	72	0.86 (0.63, 1.15)
Q 3	151,261	1.00 (1.00, 1.01)	39,235	0.96 (0.95, 0.98)	60	0.66 (0.48, 0.91)
Q 4	138,972	0.98 (0.98, 0.99)	39,993	0.93 (0.92, 0.95)	79	0.69 (0.52, 0.93)
Q 5 (highest)	132,305	0.97 (0.96, 0.98)	40,599	0.89 (0.88, 0.90)	102	0.73 (0.55, 0.95)
Comorbidities						
Hypertension	333,762	1.12 (1.11, 1.12) ^1^	125,234	1.35 (1.34, 1.37) ^1^	318	1.45 (1.11, 1.89) ^1^
Diabetes mellitus	132,914	1.12 (1.11, 1.12) ^1^	56,281	1.31 (1.30, 1.33) ^1^	162	1.59 (1.28, 1.97) ^1^
Dyslipidemia	326,615	1.04 (1.04, 1.05) ^1^	115,604	1.09 (1.08, 1.10) ^1^	236	0.67 (0.53, 0.83) ^1^
COPD	79,983	1.53 (1.52, 1.54) ^1^	35,109	1.80 (1.78, 1.83) ^1^	110	1.64 (1.30, 2.07) ^1^
Ischemic heart disease	109,276	2.56 (2.54, 2.58) ^1^	49,619	2.62 (2.59, 2.66) ^1^	147	2.35 (1.85, 3.00) ^1^
Stroke	33,480	2.09 (2.06, 2.11) ^1^	21,645	3.11 (3.06, 3.16) ^1^	64	2.54 (1.91, 3.38) ^1^
Congestive heart failure	29,766	1.65 (1.63, 1.68) ^1^	17,894	2.20 (2.16, 2.24) ^1^	91	3.91 (2.99, 5.12) ^1^
Chronic kidney disease	44,258	1.07 (1.06, 1.08) ^1^	21,816	1.19 (1.17, 1.21) ^1^	81	1.40 (1.08, 1.82) ^1^
Malignancy	30,796	1.73 (1.71, 1.75) ^1^	15,393	2.48 (2.44, 2.53) ^1^	101	6.10 (4.83, 7.71) ^1^
Depression	84,603	1.93 (1.91, 1.94) ^1^	36,097	2.13 (2.11, 2.16) ^1^	66	1.14 (0.86, 1.50) ^1^
Disability (vs. non-disability)	66,424	1.24 (1.23, 1.25)	28,274	1.31 (1.29, 1.33)	91	1.45 (1.14, 1.85)
Previous ED visit ≥ 1 (vs. none)	112,053	2.51 (2.50, 2.53)	-	-	64	1.23 (0.93, 1.62)
Previous hospitalization ≥ 1 (vs. none)	-	-	17,428	2.57 (2.52, 2.61)	-	-
BMI, kg/m^2^						
<18.5	31,639	1.18 (1.16, 1.19)	9577	1.37 (1.34, 1.40)	38	1.93 (1.35, 2.74)
18.5–22.9	273,282	1 (Ref.)	74,617	1 (Ref.)	188	1 (Ref.)
23–24.9	169,593	0.93 (0.93, 0.94)	47,971	0.88 (0.87, 0.89)	87	0.62 (0.48, 0.81)
25–29.9	212,441	0.91 (0.91, 0.92)	60,733	0.83 (0.82, 0.84)	96	0.54 (0.42, 0.69)
≥30	34,352	0.91 (0.90, 0.92)	9669	0.84 (0.82, 0.86)	13	0.55 (0.31, 0.98)
Smoking status						
Never smoked	439,152	1 (Ref.)	123,387	1 (Ref.)	247	1 (Ref.)
Ex-smoker	120,676	1.07 (1.06, 1.08)	38,494	1.11 (1.10, 1.13)	97	0.86 (0.66, 1.13)
Current smoker	161,479	1.14 (1.14, 1.15)	40,686	1.18 (1.17, 1.20)	78	1.01 (0.75, 1.37)
Alcohol consumption status						
Non-drinker	403,809	1 (Ref.)	131,055	1 (Ref.)	295	1 (Ref.)
Mild drinker	261,050	0.95 (0.95, 0.96)	57,648	0.81 (0.80, 0.82)	92	0.80 (0.62, 1.03)
Heavy drinker	56,448	1.11 (1.09, 1.12)	13,864	0.96 (0.95, 0.98)	35	1.41 (0.96, 2.06)
Regular exercise (vs. no exercise)	136,402	0.94 (0.94, 0.95)	35,984	0.87 (0.86, 0.88)	68	0.79 (0.61, 1.03)

Abbreviations: ED = emergency department, Q = quartile, COPD = chronic obstructive pulmonary disease, BMI = body mass index, aOR = adjusted odds ratio, CI = confidence interval, Ref. = reference. * Adjusted for all list variables. ^1^ Odds ratios (95% CI) relative to subjects who did not have comorbidities (malignancy, stroke, ischemic heart disease, congestive heart failure, chronic obstructive pulmonary disease, chronic kidney disease, hypertension, diabetes mellitus, dyslipidemia, depression).

## Data Availability

The data that support the findings of this study are available from the Korean National Health Insurance Service (KNHIS), which were used under license for the current study. However, restrictions apply to their availability, so they are not publicly available. From authors upon reasonable request, data are available with permission of the KNHIS.

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
