# Peer review of "Factors Associated with Emergency Department Visits and Consequent Hospitalization and Death in Korea Using a Population-Based National Health Database"

_healthcare, 2022, doi:10.3390/healthcare10071324_

Round 1
Reviewer 1 Report
Referee Report Manuscript # 1779163
Title: Factors Associated with Emergency Department Visits and Consequent Hospitalization and Death in Korea Using a Population-Based National Health Database
The authors present a thorough statistical analysis of emergency room visits in Korea. They use the official database to determine important factors in hospitalization and deaths. Although the conclusions of work show that the population at risk seems to be common sense, this data analysis might be important to determine future public health policies.
This is a well written manuscript with a good motivation of the study based on the increase of the usage of the emergency department in Korea. The methodology used is sound and the results are statistically significant, thus justifying the conclusion drawn from the work.
Given the above I suggest accepting the manuscript in its present form.
Author Response
We appreciate your valuable comments. Emergency department (ED) use has been increasing in Korea and we tried to explore the important factors associated with ED visits, hospitalization and deaths. We thought that determining the primary diagnoses and high-risk groups would be important for future public health policy.
Reviewer 2 Report
All part of manuscript were described and presented correctly. The article may be considered for publication in the Healthcare journal
Author Response
Thank you considering our manuscript for publication in Healthcare.
Reviewer 3 Report
The study aimed to investigate socioeconomic, medical, and behavioral risk factors for Emergency Department visits, hospitalization, and death in Korea using a population based national health database.
The study population is too general and consider any type of disease. All variables are analyzed individually, such as comorbidities, habits and lifestyles, without any stratification of the data, for example by type of disease, severity, type of hospitalization. The data thus expressed are not original and scientifically valid. Authors should explain in detail what the thesis of the work is, and choose a specific focus and justify the choice with a rationale. The vastness of the data analyzed without a specific focus does not allow us to draw valid conclusions.
Author Response
We agree with your comment that our study is general and that it could have been more profound with stratification of data. However, regarding the general data, although there are some studies in the US or the UK, not much research is done in Korea. We believe that exploring the factors associated with emergency department visits, consequent admissions, and mortality especially in terms of primary and diagnoses and finding high-risk groups will have significant implications for public health. Furthermore, we believe that our study could serve as a foundation for future studies with more specific aims and considerations. Thank you for your valuable comments.
Reviewer 4 Report
This manuscript studies the different factors which may contribute to the visit of the Emergency department and conseuquence afterwards in Korean population. The introduction, methods are well described and the results also clearly presented. The authors also mentioned the limitation of this study. This study could be helpful for the future consideration of medical care and polices.
There is one minor point on the understanding of the results.
For Table 4, what does the "ref" mean ? There is no definition in the table or main context.
Author Response
We appreciate your valuable comments. As you mentioned, we thought that finding different factors for emergency department visits and consequences afterwards would be helpful for future medical care and health policy.
Ref. stands for reference. We added the definition as a footnote in Table 4.
Round 2
Reviewer 3 Report
I understood the reasons expressed by the authors and I consider the manuscript to be publishable